

# The vaccina*T*ion & *H*pv *K*nowledge (THinK) questionnaire: a reliability and validity study on a sample of women living in Sicily (southern-Italy)

Domenica Matranga[1], Cristina Lumia[2], Rossella Guarneri[2], Vincenza Maria Arculeo[2], Marcello Noto[2], Alessia Pivetti[2], Gregorio Serra[2], Maria Francesca Guarneri[2] and Antonio Spera[2]

[1] Department of Health Promotion Sciences, Maternal and Infant Care, Internal Medicine and Medical Specialties "G. D'Alessandro", University of Palermo, Palermo, Sicily, Italia

[2] Unit of Gynecology and Obstetrics, Azienda Ospedaliera Universitaria Policlinico "Paolo Giaccone", Palermo, Sicily, Italia

## ABSTRACT

**Objective**. The aim of this study was to introduce the VaccinaTion & Hpv Knowledge (THinK) questionnaire to assess knowledge about human papillomavirus (HPV) and attitude to HPV-vaccination. Its reliability and validity was demonstrated in a sample of women living in Sicily (southern Italy).

**Methods**. A cross-sectional survey was conducted on a sample of 220 women at the "Paolo Giaccone" University Hospital in Palermo (Sicily), aged 18–61. Data were analyzed through Cronbach's alpha and exploratory factor analysis, followed by a structural equation model with measurement component. The two-level data structure was explicitly considered.

**Results**. Three dimensions were found: "knowledge of HPV infection (kHPV), "Attitude to be vaccinated against HPV (aHPV)" and "Knowledge about vaccines (KV)" (97% overall explained variance). Internal consistency was good for the whole questionnaire (0.82) and the first dimension (0.88) and acceptable for the second (0.78) and the third dimension (0.73). 23% of women showed no or little knowledge of HPV and 44.3% of women had no or little knowledge about HPV induced lesions.

**Discussion**. The use of a validated questionnaire may serve as a useful measure to assess general knowledge about HPV and attitude towards vaccination against HPV in the primary prevention setting.

Corresponding author
Domenica Matranga,
domenica.matranga@unipa.it

## INTRODUCTION

Human Papilloma Virus (HPV) infection is the most frequent among sexually transmitted diseases in the world. Cervical cancer is closely related to virus action, which is the main cause of cervical intraepithelial neoplasia and invasive lesions (*Fang, Zhang & Jin, 2014*).

Three HPV vaccines are at the moment available in many countries throughout the world. The bivalent (Cervarix, GSK biologicals) HPV vaccine prevents infections with the high-risk (HR) HPV 16 and 18. These genotypes are responsible for approximately 70% of cervical cancer cases globally and are considered responsible for a significant number of cervical low- and high- grade squamous intraepithelial lesions (LSIL and HSIL, respectively) (*Clifford, Rana & Franceschi, 2003*). The quadrivalent HPV vaccine (Gardasil, Sanofi Pasteur MSD), in addition to HR 16 and 18, also targets the LR HPV 6 and 11 that are associated with 90% of anogenital warts in men and women (*Braaten & Laufer, 2008*). Both bivalent and tetravalent vaccines have been shown to be effective and immunogenically valid in trials conducted in recent years with efficacy tests up to 55 years, especially in those who are virus-naïve (*Harper & DeMars, 2017*). Finally, the enavalent HPV vaccine (Merk, Sanofi Pasteur MSD, 9vHPV, trade name Gardasil9), in addition to the four genotypes of the quadrivalent vaccine, also targets five additional HR genotypes, namely HPV 31/33/45/52/58, which are the most frequently detected types in invasive cervical cancer worldwide, after HPV 16 and HPV 18 (*Capra et al., 2017*). Vaccination can be administered to people who did not have any contact with the genotypes that are covered; for this reason, it is preferable to get vaccinated in adolescence, before sexual activity begins and before any potential exposure to virus (*Loke et al., 2017*).

A cross-sectional study showed a prevalence of HR HPV infections of 24% in a group of young Sicilian women (18–24) (*Ammatuna et al., 2008*). According to the official statistics about the HPV vaccine coverage (year 2015), in Sicily the percentage of young women (fully) vaccinated against HPV is 44.10% compared to the national average rate of 62.15% (*Ministero della Salute, 2015*). In a female population living in Sicily, it has been shown that the switch to the enavalent vaccine would increase the prevention of cervical HSIL in up to 90% of cases (*Capra et al., 2017*).

Awareness of the risks associated to HPV infection is extremely important. In a large systematic review, *Hendry et al. (2013)* showed that misperception of risk could prevent from accepting vaccination; moreover, the correct knowledge of virus epidemiology can lead to adopt behaviors, as an example the use of condoms, to minimize the risk of infection. Most of the surveys conducted until now about HPV awareness and attitude to specific vaccination have involved young people (*Sopracordevole et al., 2012*; *Pelucchi et al., 2010*). In 2008 an Italian Survey among women 14–24 showed the need to strengthen HPV knowledge, since only 23.3% of interviewed have heard about HPV and cervical cancer (*Di Giuseppe et al., 2008*). Knowledge about HPV infection has been shown to be poor among the public, students, patients and health professionals (*Klug, Hukelmann & Blettner, 2008*; *Santangelo, Provenzano & Firenze, 2018*) and, more recently, among European adolescents (*Patel et al., 2016*). Furthermore, according to other studies, vaccination in women over 25 years, together with a screening program, offers the opportunity to reduce the incidence of cervical cancer in countries with limited resources and high disease burden. In 2017, results from a large survey in Italy showed that 73.8% of interviewed people were conscious about the availability of HPV vaccine, but had no trust in vaccines and believed that a PAP test is enough for prevention were expressed by 14.0% and 14.3% of women respectively (*Censis, 2018*).

The aim of this study was to assess knowledge of HPV and attitude to HPV-vaccination in a sample of Sicilian adult women and to demonstrate reliability and validity of the questionnaire used. The choice of an adult target population for this study relates to the importance of disseminating the culture of vaccination, just increasing parental awareness and attitude. The VaccinaTion & Hpv Knowledge (THinK) questionnaire was developed to be used in the first approach to the patient both in hospital and in outpatient service.

## MATERIALS & METHODS

### Participants

A cross-sectional survey was conducted at the "Paolo Giaccone" University Hospital in Palermo (Sicily) from April to December 2017. The study included 220 women, aged 18-61, consecutively enrolled from the Unit of Gynecology and Obstetrics (Ob/Gyn) (136 women) and from the University ambulatory clinic (UAC) (84 women). The Ob/Gyn is an outpatient service for women of all age groups while the UAC supplies free healthcare services to students and fresh graduates. Women apply to the Ob/Gyn either as they complain about some acute or chronic disorders or as they ask for a complete gynecological check-up. Alternatively, women apply to the UAC in order to receive information on contraception, sexually transmitted diseases (STD) and/or gynecological screening visits. Women already vaccinated against HPV were excluded from the study.

### The vaccinaTion & Hpv Knowledge (THinK) questionnaire

Enrolled women were given advices about the aim of the study. This study was conducted according to the guidelines laid down in the Declaration of Helsinki and all procedures were approved by the Research Ethics Committee at "Paolo Giaccone" University Hospital (Reference number 8/09/2018). Verbal informed consent was obtained from all participants. The THinK questionnaire included 16 items (Fig. 1), using a 5-level Likert scale (yes, much, somewhat, little, no). Questions concerned general knowledge about vaccination (acceptance, administration, effectiveness), HPV and related risks and acceptability of vaccine. The age, birthplace and education of each respondent were requested too. The draft of the THinK questionnaire was reviewed by five experts within "Paolo Giaccone" University Hospital to check its completeness and its suitability to be used in the first approach to the patient both in hospital and in outpatient service. Recommendations for improvement were also sought.

### Statistical methods

Descriptive statistics were calculated for participants' general characteristics.

As a first step, an exploratory factor analysis (EFA) was carried out to describe the joint variability of the dimensions of the THinK questionnaire. Factors were rotated using the varimax approach to ease interpretation. Internal consistency was assessed for each dimension using the Cronbach's alpha coefficient. Higher values indicate that scores on the considered dimension are internally consistent. Internal consistency is considered poor if the alpha value is below 0.60, questionable if between 0.60 and 0.70, acceptable between 0.70 and 0.80, good between 0.80 and 0.90 and excellent if not less than 0.90

1. Do you know what vaccines are?

    □ Yes
    □ Much
    □ Somewhat
    □ Little
    □ No

2. Are you favourable with paediatric vaccination?

    □ Yes
    □ Much
    □ Somewhat
    □ Little
    □ No

3. Are you favourable with adults' vaccination?

    □ Yes
    □ Much
    □ Somewhat
    □ Little
    □ No

4. Do you know what vaccines are available today for the Italian population?

    □ Yes
    □ Much
    □ Somewhat
    □ Little
    □ No

5. Do you know by who and where could you be vaccinated?

    □ Yes
    □ Much
    □ Somewhat
    □ Little
    □ No

6. Do you think that vaccines have any side effects?

    □ Yes
    □ Much
    □ Somewhat
    □ Little
    □ No

7. Can you contract a disease even if you are vaccinated against it?

    □ Yes
    □ Much
    □ Somewhat
    □ Little
    □ No

8. Do you think that vaccination is effective even after contracting infection or having been in contact with a contagious case?

    □ Yes
    □ Much
    □ Somewhat
    □ Little
    □ No

9. Do you know what HPV is?

    □ Yes
    □ Much
    □ Somewhat
    □ Little
    □ No

10. Do you think that HPV is dangerous?

    □ Yes
    □ Much
    □ Somewhat
    □ Little
    □ No

11. Do you know lesions related to HPV infection?

    □ Yes
    □ Much
    □ Somewhat
    □ Little
    □ No

12. Have you ever heard about vaccination and prevention against HPV?

    □ Yes
    □ Much
    □ Somewhat
    □ Little
    □ No

13. Do you think that is high the probability of contracting HPV infection?

    □ Yes
    □ Much
    □ Somewhat
    □ Little
    □ No

14. Would you be willing to get vaccinated against HPV?

    □ Yes
    □ Much
    □ Somewhat
    □ Little
    □ No

15. Do you consider useful asking to your partner to get vaccinated against HPV?

    □ Yes
    □ Much
    □ Somewhat
    □ Little
    □ No

16. Do you want to receive information about HPV vaccination?

    □ Yes
    □ Much
    □ Somewhat
    □ Little
    □ No

**Figure 1** **The 16-items THinK questionnaire.**

(*Bland & Altman, 1997*). The sample size for this study was calculated according to the rule required in internal validity studies, which uses the ratio of the number of subjects (N) to the number of items (p) (*Rouquette & Falissard, 2011*).

**Table 1  Descriptive statistics of 220 Sicilian women by recruitment group.**

| Variables | Ob/Gyn Department (N = 136) | University outpatient service (N = 84) | p-value[a] |
|---|---|---|---|
| Age (Mean ± SD) | 35.50 ± 9.89 | 23.12 ± 2.45 | <0.001 |
| Education (n,%) | | | <0.001 |
| No, primary | 1 (0.8) | 0 (0.0) | |
| Low Middle School | 49 (36.6) | 0 (0.0) | |
| High Middle School | 49 (36.6) | 70 (83.3) | |
| Graduate | 35 (26.0) | 14 (16.7) | |
| Citizenship (n,%) | | | <0.001 |
| Italian | 131 (96.3) | 80 (95.2) | 0.830 |
| European | 2 (1.5) | 3 (3.6) | |
| Extra-European | 2 (1.5) | 1 (1.2) | |
| n.a. | 1 (0.7) | – | |
| Living place (n,%) | | | |
| Italy | 134 (98.6) | 81 (96.4) | 0.213 |
| Europe | 1 (0.7) | 3 (3.6) | |
| Extra-Europe | – | – | |
| n.a. | 1 (0.7) | – | |

**Notes.**

[a]Student's *t*-test for quantitative variables, Chi$^2$ or Fisher's exact test for categorical variables

As a second step, a structural equation model with measurement component was estimated to confirm the factor structure obtained through EFA. It was assumed a generalized linear model for ordinal response and link logit. As a final step, in order to take into account that participants are nested within two groups Ob/Gyn and UAC, a two-levels measurement model was considered. The LR test, AIC and BIC were used for comparison between the one-level and two-levels SEM models. The Student's t-test was used to assess statistical significance of the difference between two women's groups with respect to questionnaire's items and dimensions. A $p$-value $< 0.05$ was chosen for statistical significance. Statistical analysis was conducted using Stata SE/14.2.

## RESULTS

All five experts concurred that the measures should effectively capture any changes in the knowledge about HPV and attitude to vaccination and HPV vaccines, with concern to the first approach to the patient both in hospital and in outpatient service.

Women enrolled at UAC were on average younger (23.1 ± 2.45) and more educated (100% high school and more) than women enrolled at Ob/Gyn (35.5 ± 9.89 years old and 63% high school and more) (Table 1).

Three dimensions were found trough EFA: "knowledge of HPV infection (kHPV)" (48% explained variance), correlated with items between Q9 and Q13, "Attitude to get vaccinated against HPV (aHPV)" (26%), correlated with items Q14-Q15-Q16, and "Knowledge about vaccines (KV)" (23%), correlated with items from Q1 to Q5. Three items (Q6-Q7, Q8) resulted with high uniqueness (≥0.80). Internal consistency was good for the whole
Table 2 Internal consistency of the THinK questionnaire.

|  | Cronbach's alpha |
| --- | --- |
| Whole questionnaire | 0.816 |
| kHPV | 0.882 |
| aHPV | 0.784 |
| KV | 0.732 |

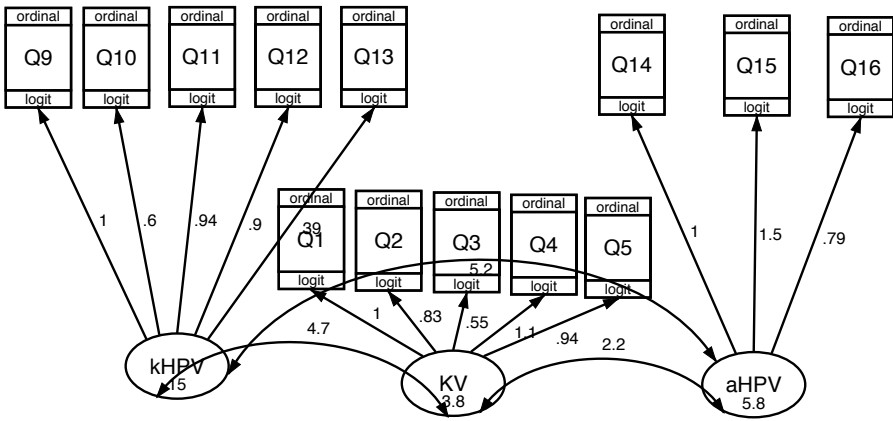

**Figure 2 Construct validity of the THinK Questionnaire in a sample of Sicilian adult women: paths obtained from 1-level measurement SEM model.** kHPV, "knowledge of THinK infection", KV, "Knowledge about vaccines", aHPV, "Attitude to get vaccinated against HPV", Q1-Q16, Questionnaire's items.

questionnaire (0.82) and the first dimension (0.88) and acceptable for the second (0.78) and the third dimension (0.73) (Table 2).

Results of the 1-level measurement SEM model confirmed the three factors structure of the THinK questionnaire (Fig. 2). By including the group's level information (Fig. 3), the model fit was improved ($p = 0.0223$).

Notably, 23.1% of responders showed poor knowledge of HPV and 16.7% had never heard of it (Q12); For what concerns specific immunization, 21.8% of patients (Q14) responded that they are very little or not at all available to carry out vaccination. Poor knowledge of HPV induced lesions was expressed by almost half of the women interviewed (44.3% of women answered no or little at Q11), without any correlation with age or education. About the 19% of women expressed a clear refusal just to receive even simple information about HPV vaccination (Q16) (data not shown in tables).

On average, women enrolled at the UAC reported scores significantly higher compared to women enrolled at Ob/Gyn for Q3 ($2.49 \pm 1.41$ vs $1.95 \pm 1.12$), Q14 ($2.60 \pm 1.70$ vs $2.14 \pm 1.20$), Q15 ($2.96 \pm 1.79$ vs $2.44 \pm 1.28$) and aHPV ($0.17 \pm 1.27$ vs $-0.28 \pm 0.93$). Conversely, their score was significantly lower for Q6 ($2.96 \pm 1.41$ vs $3.57 \pm 1.14$) and Q7 ($3.15 \pm 1.15$ vs $3.68 \pm 1.14$) (Table 3).
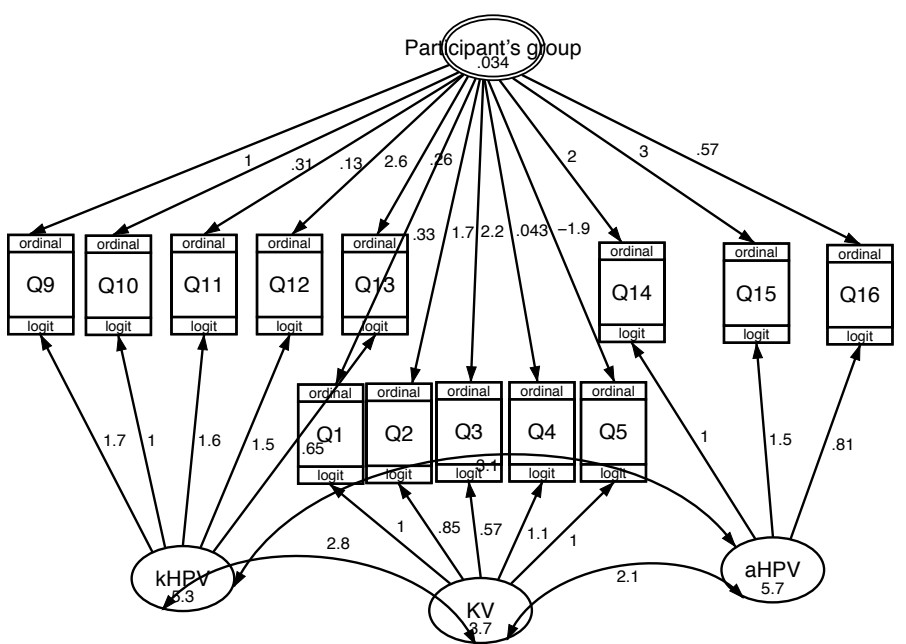

**Figure 3 Construct validity of the THinK Questionnaire in a sample of Sicilian adult women: paths obtained from two-levels measurement SEM model.** kHPV, "knowledge of HPV infection", KV, "Knowledge about vaccines", aHPV, "Attitude to get vaccinated against HPV", Q1-Q16, Questionnaire's items.

## DISCUSSION

The aim of our study was to perform a reliability and validity study of the proposed THinK questionnaire as well as to assess knowledge of HPV and attitude towards HPV-vaccination in a sample of women living in Palermo.

The THinK questionnaire was found to measure three domains, which we named knowledge of HPV infection, attitude to be vaccinated against HPV and knowledge about vaccines, and it demonstrated adequate internal consistency as a whole and for each one of its three domains. We developed a short and succinct questionnaire with only 16 items in order to get it easier to be used in the first approach to the patient both in hospital and in outpatient service. There are many studies employing questionnaires about HPV issues, but just few use validated questionnaires, and their validity results are in line with those found for the THinK questionnaire. The HAVIQ questionnaire (*Forster et al., 2017*), which was designed to evaluate the efficacy of an intervention to affect HPV vaccination knowledge, obtained Cronbach's alpha >0.6 for three of its four dimensions. The 25-item HPV general knowledge and 11-item HPV vaccination scale, validated on a national sample of Canadian parents of boys, showed internal consistency >0.70 (*Perez et al., 2016*). Two questionnaires to measure HPV knowledge on an international sample of adults showed Cronbach's alpha >0.83 (*Waller et al., 2013*). Similar reliability was demonstrated for other instruments validated on undergraduates from Pakistan (Cronbach's alpha (0.79)) (*Khan et al., 2016*) and on adolescents from Greece (*Anagnostou, Aletras & Niakas, 2017*).

**Table 3** Knowledge and attitude of 220 Italian women by group: Mean (SD) of responses to THinK questionnaire's items and dimensions.

| Questionnaire's items | Ob/Gyn Department (N = 136) | University outpatient service (N = 84) | p-value |
|---|---|---|---|
| $Q_1$ | 1.74 (1.01) | 1.75 (0.99) | 0.9310 |
| $Q_2$ | 1.55 (0.95) | 1.83 (1.15) | 0.0589 |
| $Q_3$ | 1.95 (1.12) | 2.49 (1.41) | **0.0019** |
| $Q_4$ | 2.92 (1.21) | 2.91 (1.23) | 0.9770 |
| $Q_5$ | 2.58 (1.25) | 2.24 (1.37) | 0.0662 |
| $Q_6$ | 3.57 (1.14) | 2.96 (1.41) | **0.0066** |
| $Q_7$ | 3.68 (1.14) | 3.15 (1.15) | **0.0029** |
| $Q_8$ | 3.58 (1.30) | 3.93 (1.44) | 0.0705 |
| $Q_9$ | 2.63 (1.32) | 2.75 (1.47) | 0.5195 |
| $Q_{10}$ | 2.30 (1.06) | 2.50 (1.61) | 0.2619 |
| $Q_{11}$ | 3.06 (1.43) | 3.09 (1.59) | 0.8929 |
| $Q_{12}$ | 2.58 (1.42) | 2.99 (1.64) | 0.0597 |
| $Q_{13}$ | 2.63 (1.07) | 2.75 (1.51) | 0.4966 |
| $Q_{14}$ | 2.14 (1.20) | 2.60 (1.70) | **0.0198** |
| $Q_{15}$ | 2.44 (1.28) | 2.96 (1.79) | **0.0138** |
| $Q_{16}$ | 2.14 (1.24) | 2.38 (1.70) | 0.2304 |
| Dimensions | | | |
| kHPV | −0.01 (0.98) | 0.00 (1.16) | 0.9412 |
| KV | −0.06 (1.24) | 0.04 (1.19) | 0.5699 |
| aHPV | −0.28 (0.93) | 0.17 (1.27) | **0.0026** |

**Notes.**
Bold values denote statistical significance at the $p < 0.05$ level.

The context proposed in our analysis concerns adult females living in Palermo. Rationale for choosing this target was to have indications about women that are yet (or not yet) mothers of girls and boys in vaccination age. Parents' attitude about immunization is extremely important to vaccination of younger girl (*La Torre et al., 2015*); disagreement of parents has been the main reason for non-adherence to vaccination of girls aged 11–12 in an Italian Cross sectional study (*Gualano et al., 2016*).

When analyzing results of our paper, it is necessary to keep in mind that two groups involved in the survey are representative of two different populations. In fact, participants from Ob/Gyn are adult women referred for therapeutic purposes while those from the UAC are students or fresh graduates mostly referred for prevention and contraceptive prescription. Results from the 2-levels measurement SEM model showed that the group information explains a significant share of the total variability of the responses. This finding strengthens the importance of validating the questionnaire to measure HPV knowledge in specific populations.

One important finding of our study is that more than one in three of women were not conscious about HPV prevention and one in five would be only a little or not willing to be vaccinated against HPV. The roots of these disappointing results can be explained by a general poor knowledge of HPV (*Hendry et al., 2013*; *Sopracordevole et al., 2012*; *Pelucchi et*

*al., 2010*; *Di Giuseppe et al., 2008*), lack of information on HPV vaccination given by health professionals to young women (*La Torre et al., 2013*) and widespread ignorance about HPV-specific lesions (*Capogrosso et al., 2015*), which has also been reported for the Italian general practitioners (GPs) (*Signorelli et al., 2014*). It has already been suggested that lack of information could represent an important barrier to vaccine acceptation (*Loke et al., 2017*); a cross-sectional pilot study showed higher vaccine-related knowledge in women vaccinated than in non-immunized (*Mathur, Mathur & Reichling, 2010*).

Our results are consistent with Censis data, which show that judgment on available information on HPV and vaccinations is not positive, in terms of clarity and quantity (*Censis, 2018*).

By evaluating answers to the last question, it is possible to hypothesize a strong cultural resistance to vaccination and let us to reflect that a rooted resistance to immunization could play an important role in the HPV vaccine hesitancy (*Guzzetta et al., 2014*). Comparison between response to our questions Q3 and Q4 appears interesting, with 18.6% of respondents declaring that they were little or not at all favorable to vaccination in adulthood and only 8.6% being little or nothing convinced of vaccination in pediatric age. The found resistance to vaccination in adulthood, compared to the consolidated acceptance of vaccination in pediatric age, should help to explain the scarce attitude to be vaccinated against HPV in the Ob/Gyn group. Barriers to vaccination in adulthood have been long discussed by several authors and one of the most frequent reasons for failure to immunize is a lack of communication or bad information (*Johnson, Nichol & Lipczynski, 2008*).

Different authors showed the impact of socioeconomic variables on HPV awareness and vaccination: a few reported positive relationship between school education and mother's school level, HPV knowledge and vaccination (*Grandahl et al., 2017*), between age and HPV awareness (*Samkange-Zeeb, Mikolajczyk & Zeeb, 2013*) and between education and the intention to be vaccinated against HPV (*Alberts et al., 2017*). Concerning the association with education, findings of our study do not allow to draw definitive conclusions, as it is not possible to distinguish if the major attitude of women enrolled at UAC depends on either the younger age or the higher education of this group compared to Ob/Gyn.

Through this study, we have provided initial evidence for validation of the THinK questionnaire. There are some margins for improvement of the instrument, as deleting those items resulted with high uniqueness or including additional items to get it more HPV-specific, e.g., concerning sexual behaviors (age at first sexual relationship, number of life partners, partner HPV status), knowledge about HPV induced lesions, smoking cigarettes, alcohol consumption, use of hormonal contraceptives or IUD, personal hygiene, prior infections of the cervico-vaginal tract, parity, HSV. Finally, more research is desirable to examine other aspects as concurrent validity and test-retest reliability with larger sample sizes.

## CONCLUSIONS

The THinK questionnaire demonstrated adequate reliability and validity in a sample of Italian women living in Palermo. This instrument, short and easy to complete and to score, may serve as a useful measure in the primary healthcare setting in order to assess general

knowledge about vaccination and HPV vaccines. Even if, in agreement with the guidelines of the Italian Ministry of Health, HPV vaccination is offered free of charge to girls in the twelfth year of life in all Italian regions since 2007/2008, efforts must be made to create a sound basis for understanding HPV issues and related risks, with a view to preventing and protecting patients.

### Funding
The authors received no funding for this work.

### Competing Interests
The authors declare there are no competing interests.

### Author Contributions
- Domenica Matranga conceived and designed the experiments, analyzed the data, contributed reagents/materials/analysis tools, prepared figures and/or tables, authored or reviewed drafts of the paper, approved the final draft.
- Cristina Lumia conceived and designed the experiments, analyzed the data, prepared figures and/or tables, authored or reviewed drafts of the paper, approved the final draft.
- Rossella Guarneri, Vincenza Maria Arculeo, Marcello Noto, Alessia Pivetti, Gregorio Serra and Maria Francesca Guarneri, performed the experiments, approved the final draft.
- Antonio Spera conceived and designed the experiments, analyzed the data, authored or reviewed drafts of the paper, approved the final draft.

### Human Ethics
The following information was supplied relating to ethical approvals (i.e., approving body and any reference numbers):
   The Research Ethics Committee at Paolo Giaccone University Hospital approved this research

### Data Availability
   The raw data are provided in the Supplemental File.

### Supplemental Information
Supplemental information for this article can be found online at http://dx.doi.org/10.7717/peerj.6254#supplemental-information.

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
