# Peer review of "The vaccinaTion & Hpv Knowledge (THinK) questionnaire: a reliability and validity study on a sample of women living in Sicily (southern-Italy)"

_PeerJ, doi:10.7717/peerj.6254_

## Round 0.1 · original submission · Minor Revisions

The manuscript recieved favorable reviews from the reviewers. However, there are still some aspects that needs further revision by authors.

·

Basic reporting

English used throughout the manuscript was clear and unambiguous, professional


The literature references describe a partial evidence produced in Italy on this issue. I suggest to consider these papers both for the introduction and the Discussion sections

Napolitano F, Napolitano P, Liguori G, Angelillo IF. Human papillomavirus infection and vaccination: Knowledge and attitudes among young males in Italy. Hum Vaccin Immunother. 2016 Jun 2;12(6):1504-10
Santangelo OE, Provenzano S, Firenze A. Knowledge of sexually transmitted infections and sex-at-risk among Italian students of health professions. Data from a one-month survey. Ann Ist Super Sanita. 2018 Jan-Mar;54(1):40-48.
La Torre G, De Vito E, Ficarra MG, Firenze A, Gregorio P, Miccoli S, Giraldi G, Unim B, De Belvis G, Boccia A, Saulle R; HPV Collaborative Group, Semyonov L, Ferrara M, Langiano E, Capizzi S, Nardella R, Marsala MG, Bonanno V, Ferrara C, Guidi E, Bergamini M, Lupi S. Knowledge, opinions and attitudes of Italian mothers towards HPV vaccination and Pap test. Tumori. 2015 May-Jun;101(3):339-46
Guzzetta G, Faustini L, Panatto D, Gasparini R, Manfredi P. The impact of HPV female immunization in Italy: model based predictions. PLoS One. 2014 Mar 11;9(3):e91698.
La Torre G, De Vito E, Ficarra MG, Firenze A, Gregorio P, Boccia A; HPV Collaborative Group. Is there a lack of information on HPV vaccination given by health professionals to young women? Vaccine. 2013 Oct 1;31(42):4710-3
Pelullo CP, Di Giuseppe G, Angelillo IF. Human papillomavirus infection: knowledge, attitudes, and behaviors among lesbian, gay men, and bisexual in Italy. PLoS One. 2012;7(8):e42856
Sopracordevole F, Cigolot F, Gardonio V, Di Giuseppe J, Boselli F, Ciavattini A. Teenagers' knowledge about HPV infection and HPV vaccination in the first year of the public vaccination programme. Eur J Clin Microbiol Infect Dis. 2012 Sep;31(9):2319-25
Pelucchi C, Esposito S, Galeone C, Semino M, Sabatini C, Picciolli I, Consolo S, Milani G, Principi N. Knowledge of human papillomavirus infection and its prevention among adolescents and parents in the greater Milan area, Northern Italy. BMC Public Health. 2010 Jun 28;10:378

The structure, figures and tables are professionally edited. Raw data are shared.
The structure of the article has an acceptable format of ‘standard sections’
Figures 1 and 2are relevant to the content of the article, of sufficient resolution, and appropriately described and labeled.

Appropriate raw data has been made available, in accordance with PeerJ Data Sharing policy.
Relevant results are related to hypotheses formulated in the aim of the study.

Experimental design

The object of the primary research is original, since according to my knowledge, there is no validated questionnaire to study the specific issue raised by the authors

The research question is clearly define, relevant and meaningful. The knowledge gap being investigated is identified in the Introduction section
The investigation has been conducted rigorously and to a high technical standard.
The authors used a cross-sectional approach, and this is correct to answer their research question.
Methods are described with sufficient information to be reproducible by another investigator. In the statistical paragraph the authors are requested to give a reference for justifying the Internal consistency.

Validity of the findings

The findings are interesting, and the authors at the end of the manuscript clearly state there is the need to replicate the study with a larger sample for test-retest reliability issue.

Data is robust, statistically sound, & controlled.
The data on which the conclusions are based are provided in an Excel file. I made my own analysis with SPPS. While I found 0.882 for the first dimension (between Q9 and Q13), 0.784 for the second dimension, and 0.732 for the last dimension, for the whole for the whole questionnaire I found a value of 0.816. Please check your results carefully.
In the discussion, clearly state the impact of the use of this questionnaire from a public health perspective.
Conclusion are well stated. However, as already said above, some references at the national level are lacking.

Additional comments

This is a very interesting paper concerning the validity and reliability of a questionnaire in measuring HPV knowledge and HPV vaccination. The paper is of interest for an international reader and could be useful to other researchers involved on this topic.
Some minor amendments are required, as stated above and following also the following points:
- Line 72-74. The authors stated “Data have been collected through a cross-sectional survey conducted at the “Paolo Giaccone” University Hospital in Palermo (Sicily).”. this statement must be moved to the Methods section
- In the Methods the authors need to describe how and when the participants were selected. Sample size calculations are not reported. If the authors think these are not applicable in this case, please explain in details.

Reviewer 2 ·

Basic reporting

The manuscript is interesting and deal with an important topic of public health. The aim of the study is clear. Discussion and conclusion are related to the aim of the article and are well structured. The English used is simple but clear.

Although there are few comments and suggestions I would like to do.
In the abstract, section method, the age of women interviewed is not specified.
In the introduction the author talks about "bivalent, tetravalent and "enavalent" vaccine (at lines 45, 47, 50), I suggest to give a little introduction of these vaccines and specify the genotype of HPV addressed. The reason why the survey is conducted among women over 25 should be clarified (lines 63-65).

Experimental design

This primary research article is in line with aims and scope of the journal.
Research question is well defined, although the aim is to assess knowledge and attitude not awareness. This consideration is for both aim of the study and the method section.
Materials and methods is complete and described with sufficient details. The reliability and validity is properly evaluated.

Validity of the findings

Data are good and well presented.
In the results a table about socio-demographic characteristics is missing. It would be better to give more explanations about the enrollment of women aged above 25 years, considering the vaccine is recommended only at a young age before sexual activity and exposure to virus. It could be added consideration about the importance of a correct culture about vaccine among the whole population that influence at a different level parent's and adolescent choice.
It would be useful to report the internal consistency of the whole questionnaire and single sub scales in a table.

Additional comments

The article is mainly well-structured, clear and correct in all its sections.

---

## Round 0.2 · accepted · Accept

The atithors have addressed the comments raised by the reviewers and the manuscript can be accepted for publication in its current format